# *Daphne kiusiana* Crude Extract and Its Fraction Enhance Keratinocyte Migration via the ERK/MMP9 Pathway

**DOI:** 10.3390/cimb47050300

**Published:** 2025-04-25

**Authors:** Jinu Lee, Seon Min Oh, Hyung Won Ryu, Jeong-Hwa Baek

**Affiliations:** 1Department of Molecular Genetics, School of Dentistry and Dental Research Institute, Seoul National University, Seoul 08826, Republic of Korea; moonsoul7@snu.ac.kr; 2Natural Product Research Center and Natural Product Central Bank, Korea Research Institute of Bioscience and Biotechnology, Cheongju-si 28116, Republic of Korea; seonmin88@kribb.re.kr (S.M.O.); ryuhw@kribb.re.kr (H.W.R.)

**Keywords:** *Daphne kiusiana*, phytomedicine, wound healing, keratinocyte migration, ERK signaling

## Abstract

*Daphne kiusiana* is a naturally occurring plant in East Asia belonging to the Thymelaeaceae family. While its biological properties have been explored, its potential role in wound healing remains largely unknown. This study investigated the effects of *Daphne kiusiana* extracts on keratinocyte migration and the underlying mechanisms. The crude extract and its fractions was tested in vitro at various concentrations to evaluate their ability to promote keratinocyte migration, and a cytotoxicity assay was conducted to assess cell viability. The results demonstrated that *Daphne kiusiana* significantly enhanced keratinocyte migration without inducing notable cytotoxicity at tested concentrations. Mechanistically, this effect was mediated through the modulation of matrix metalloproteinase-9 (MMP-9) via extracellular signal-regulated kinase (ERK) signaling, which plays a crucial role in keratinocyte migration. These findings suggest that *Daphne kiusiana* may serve as a potential therapeutic agent for enhancing keratinocyte migration in wound healing. However, further investigations, including clinical studies, are necessary to confirm its efficacy and safety. To our knowledge, this is the first study to report the potential of *Daphne kiusiana* in promoting keratinocyte migration, offering new insights into wound healing therapies.

## 1. Introduction

Wound healing is a complex biological process that encompasses multiple phases, including hemostasis, inflammation, cell proliferation and migration, and tissue remodeling. These phases overlap and harmonize for the successful completion of wound healing [1]. Among these, the migration of keratinocytes from the wound margins into the wound bed following the inflammatory phase is a pivotal step for re-epithelialization and the prevention of pathogen infections. Impairments in keratinocyte migration are commonly associated with chronic wounds, such as diabetic foot ulcers, in which prolonged ulceration significantly increases the risk of amputation [2,3]. Given these considerations, novel approaches are needed to enhance keratinocyte migration to improve wound resolution.

Various approaches have been explored to promote wound healing under challenging circumstances. For example, becaplermin gel remains the only FDA-approved ointment or gel-type drug that directly targets diabetic foot ulcers [4]. Other treatment methods, including physical therapy and antimicrobial agents, are primarily limited to improving the local wound environment. Furthermore, conventional interventions still have limitations. For example, antimicrobial reagents exhibit toxicity at high doses [5]. Therefore, it is necessary to identify potential therapeutic agents that can directly enhance key cellular processes involved in wound repair, such as keratinocyte migration.

Accordingly, interest in natural extracts has been steadily increasing, and many studies highlight their significant potential for wound healing due to their diverse bioactive compounds [6]. Notably, Sidhu et al. elucidated the beneficial impacts of curcumin on wound repair [7]. Moreover, understanding the mechanisms through which natural extracts modulate the wound healing process is crucial for the refinement of their therapeutic application. In this context, Wahedi et al. have reported the capacity of Aloesin, an extract from *Aloe vera*, to influence Mitogen-Activated Protein Kinase (MAPK)/Rho signaling pathways pivotal for cellular migration [8].

Thus, the present study focuses on exploring the utility of *Daphne kiusiana* from the family of Thymelaeaceae. *Daphne kiusiana* has been reported to contain phenolic and flavonoid compounds, which may be responsible for antioxidant activity [9]. Furthermore, *Daphne kiusiana* alleviated inflammation in the neuronal and pulmonary systems [10,11,12]. However, its wound healing effects have not yet been explored. Thus, this study aims to investigate the effects of *Daphne kiusiana* on the cell migration stage of the wound healing process, elucidating the cellular and molecular mechanisms governing its therapeutic effects.

## 2. Materials and Methods

### 2.1. Plant Material

*Daphne kiusiana* Miquel is an evergreen broadleaf shrub found in abundance in the southern part of the Republic of Korea and Jeju Island. A specimen of *Daphne kiusiana* was collected and dried in 2019, as reported in the previous literature, and was used in this study (original reference specimen: NIBRVP0000726155) [12]. Of the collected aerial parts, only stems were used to obtain a better yield.

### 2.2. Preparation of Daphne kiusiana Stem Extracts

The extracts and fractions were separated from a dried stem of *Daphne kiusiana,* as previously described [12]. Briefly, the total *Daphne kiusiana* stem extract (DKS, yield 9.3%) was separated using SPOT-II MPLC (medium-pressure liquid chromatography) (Gilson, Middleton, WI, USA). Reversed-phase silica gel (YMC-Pack ODS-AQ HG, 20 × 250 mm, 10 µm, Kyoto, Japan) was used for elution with a mobile phase consisting of water (solvent A) and methanol (solvent B). The gradient elution conditions were established as follows: 0–10 min, 10% B; 10–70 min, 70% B; 70–71 min, 100% B; and 71–90 min, maintained at 100% B. This method resulted in the separation of DKS Nos. 1–6 (see Figure 1A). Six compounds were tentatively identified from fraction No. 5 of the DKS (DKF5), which was found to be the most active fraction, using UPLC-QToF/MS. The analyzed compounds were identified as daphnodorin I (14), nortrachelogenin (15), daphnodorin B (16), daphnodorin D2 (17), daphnodorin C (18), daphnodorin F (19), daphnodorin E (20), daphnodorin A (21), and daphneolone (22). The structures of the compounds were determined using spectroscopic data, primarily from an *in-house* library, along with retention times (RT), *m*/*z* values, molecular formulas, and mass errors (ppm) based on previously published data [12].

### 2.3. UPLC-QToF/MS Conditions

Waters Vion IMS Q-TOF mass spectrometer (Waters Corp., Milford, MA, USA) was coupled with the ACQUITY UPLC I-Class system. The analytical column was analyzed using an ACQUITY UPLC BEH C18 reversed-phase column (100 × 2.1 mm, 1.7 µm, Waters). The solvent composition comprised water containing 0.1% formic acid (solvent A, *v*/*v*) and acetonitrile containing 0.1% formic acid (solvent B, *v*/*v*). The analytical gradient methods were B 5% (0 min–1 min) → 100% (20 min–22.3 min) → 5% (22.4 min–25 min). The flow rate was eluted at 0.4 mL/min, the temperature of the column was set to maintain 35 °C, and the sample injection volume was 2 µL. The QToF/MS analysis was performed in electrospray ionization (ESI), with ion source temperature at 110 °C and desolvation temperature at 350 °C. The capillary voltage was analyzed at 2.3 kV and the cone voltage at 40 V, and the desolvation and cone gas flow rates were set to 800 L/h and 50 L/h, respectively. Accurate mass measurements were obtained with an automatic calibrated delivery system using leucine-enkephalin as the internal reference solution [*m*/*z* 554.2615 (ESI−), *m*/*z* 556.2771 (ESI+)]. The mass value scan was set to the range of *m*/*z* 100–1500. All MS data were analyzed in MSE mode, and mass spectrometry data were acquired using UNIFI software (version 1.9.4.053, Waters).

### 2.4. Reagents

Dulbecco’s modified Eagle’s medium (DMEM) and fetal bovine serum (FBS) were purchased from Hyclone (Logan, UT, USA). Easy-BLUE^TM^ and PRO-PREP^TM^ were obtained from iNtRON Biotechnology (Sungnam, Republic of Korea). For first-strand cDNA synthesis, AccuPower RT Premix was purchased from Bioneer (Daejeon, Republic of Korea), and TB Green Premix Ex Taq was obtained from Takara (Shiga, Japan). The PCR primers were synthesized from Cosmogenetech (Seoul, Republic of Korea). Antibodies for ERK and phospho-ERK were purchased from Cell Signaling (Danvers, MA, USA), and the β-actin antibody was purchased from Santa Cruz (Dallas, TX, USA). Mitomycin C was purchased from Sigma Aldrich (Burlington, MA, USA). EZ-Cytox for cell viability assays was purchased from DoGenBio (Seoul, Republic of Korea). The selective inhibitor of MEK1/2, U0126, was obtained from Merck (Rahway, NJ, USA), and the MMP9 inhibitor, ab142180, was purchased from Abcam (Cambridge, UK).

### 2.5. Cell Culture

HaCaT cells, spontaneously immortalized human keratinocytes, were obtained from Professor Kyung Mi Woo at Seoul National University. HaCaT cells were maintained in high-glucose DMEM supplemented with 10% heat-inactivated FBS and antibiotics (100 U/mL penicillin and 100 µg/mL streptomycin) at 37 °C in 5% CO_2_ incubator.

HaCaT cells were exposed to DKS (40 µg/mL) or DKF5 (1 µg/mL) for 24 or 48 h with or without pretreatment with U0126 (10 µM) or ab142180 (10 µM) 1 h prior. After the indicated time points, cells were collected for the subsequent experiments.

### 2.6. Cell Viability Assay

HaCaT cells were seeded onto 96-well plates at a density of 1 × 10^4^ cells/well and cultured to 80% confluency. The cells were incubated with the indicated concentrations of DKS or DKF5 for 24 or 48 h. Then, 10 µL of EZ-Cytox was added into each well and incubated at 37 °C in 5% CO_2_ incubator for 2 h. The absorbance was measured at 450 nm using a multiplate reader, GlowMax (Promega, Madison, WI, USA).

### 2.7. In Vitro Scratch Test

Cells were cultured in four-well plates (Thermo Fisher, Waltham, MA, USA) and grown to confluence at 37 °C in a 5% CO_2_ incubator. The cells were serum-starved for 24 h, then treated with mitomycin C (5 µM) in serum-free DMEM for 2 h. Following two washes with PBS, a scratch was created by manually scraping the cell monolayer with a 200 µL pipette tip. Subsequently, cells were incubated with the indicated concentrations of DKS or DKF5 with or without U0126 or MMP9 inhibitor at the specified concentrations. The wound area was measured using photomicrography (Olympus, Tokyo, Japan) to compare the remaining wound area at 0, 24, and 48 h after the initial scratch at 200× magnification. Images of the scratched areas were analyzed using ImageJ software 1.53e.

### 2.8. Transwell Migration Assay

HaCaT cells were incubated in serum-free conditions overnight. After that, cells were detached and seeded into the upper inserts (8.0 µm pore size) at a density 1 × 10^5^ cells/well in 100 µL of DMEM in Transwell migration assay plates (Corning Inc., Corning, NY, USA). The lower chamber was filled with 600 µL DMEM containing 1% FBS with the indicated concentrations of DKS, DKF5, U0126, or ab142180 for 24 h. Migrated cells adhering to the lower membrane of the insert were subsequently stained with DAPI, and the number of cells was quantified to assess migration capability.

### 2.9. Quantitative Reverse Transcription-Polymerase Chain Reaction (qRT-PCR)

Total RNA was isolated from cells using easy-BLUE™ RNA extraction reagent, and complementary DNA was synthesized from 2 µg of total RNA using AccuPower RT PreMix. qRT-PCR was conducted using TB Green Premix Ex Taq on 7500 Real-Time PCR system (Applied Biosystem, Foster City, CA, USA). Levels of the gene expression were compared after normalization to internal control, human glyceraldehyde-3-phosphate dehydrogenase (GAPDH). The human primer sequences used for qRT-PCR were as follows: *GAPDH*, forward 5′-CCA TCT TCC AGG AGC GAG ATC-3′ and reverse 5′-GCC TTC TCC ATG GTG GTG AA-3′; *MMP9*, forward 5′-GCA CAA CGT CTT CCA GTA-3′ and reverse 5′-GCA CTG CAG GAT GTC ATA-3′.

### 2.10. Western Blot

Cells were harvested at the end of the culture period and lysed using PRO-PREP™. After sonication, samples containing equal amounts of protein were subjected to sodium dodecyl sulfate–polyacrylamide gel electrophoresis (SDS–PAGE) and subsequently electro-transferred onto a polyvinylidene difluoride membrane. The membrane was blocked with 5% non-fat dry milk or 5% bovine serum albumin in Tris-buffered saline containing 0.1% Tween 20 for 1 h at room temperature, then incubated with the indicated primary antibody, and followed by incubation with an HRP-conjugated secondary antibody. Immune complexes were visualized using Sensi-view^TM^ Pico ECL Reagent (Lugen Sci. Inc., Bucheon, Republic of Korea) and detected using a MicroChemi imaging system (DNR, Jerusalem, Israel).

### 2.11. Zymography

After the treatment of DKS or DKF5 to HaCaT cells for 24 or 48 h, the supernatants were collected. Supernatant samples were loaded onto a 10% SDS–PAGE gel containing 0.1% gelatin for electrophoresis. After an electrophoretic run, the gel was washed with 2.5% Triton X-100 (*w*/*v*) at room temperature for 30 min and incubated overnight in a developing buffer at 37 °C. The gel was stained for 1 h with 0.5% Coomassie blue and then destained with 10% methanol and 5% acetic acid at room temperature. Transparent bands indicating the gelatinolytic activity of MMPs were visualized against a blue background (gel retaining stain). Gels were imaged using a digital imaging system. The detailed protocol was adapted from Frankowski et al. [13].

### 2.12. Statistical Analysis

All the quantitative data are presented as mean ± standard deviation (SD). Statistical significance was determined using Student’s *t*-test or one-way ANOVA with Tukey post hoc analysis performed in Prism5. A *p*-value < 0.05 was considered statistically significant.

## 3. Results

### 3.1. Tentative Identification of Main Compounds

The UPLC-QToF/MS chromatograms showed that peaks of *Daphne kiusiana* stem extract (DKS) and fraction 5 (DKF5) ranged from 5.5 to 9.0 min (Figure 1A,B). It was found that five main peaks were commonly present in fraction 5 (Figure 1C). In particular, they were abundant in MS analyses conducted using a reversed-phase column with a gradient of polar solvents. However, since these compounds share the same molecular weight and molecular formula, standard compounds must be used to distinguish between the peaks accurately. Experimental structural identification was conducted to select five major components with high content at 254 nm for target components predicted to be biflavonoids, with a focus on tentatively identifying the phytochemical substances.

As shown in Table 1, compound number 14 was identified as a flavonoid based on its adduct ion at *m*/*z* 541 [M−H]^−^ and *m*/*z* 1083 [2M−H]^−^. The molecular formula was confirmed as C_30_H_22_O_10_ using the chemical composition function of the Waters UNIFI program. Referring to previous papers, it was tentatively inferred to be the compound daphnodorin I [14]. Compound 15 was identified as lignan based on its adduct ion at *m*/*z* 373 [M−H]^−^. The molecular formula was confirmed as C_20_H_22_O_7_ using the chemical composition function of the Waters UNIFI program. Referring to previous papers, it was tentatively inferred to be the compound nortrachelogenin [15]. Compound 16 was identified as a flavonoid based on its adduct ion at *m*/*z* 541 [M−H]^−^ and *m*/*z* 1083 [2M−H]^−^. The molecular formula was confirmed as C_30_H_22_O_10_ using the chemical composition function of the Waters UNIFI program. Referring to previous papers, it was tentatively inferred to be the compound daphnodorin B [16]. Compound 18 was identified as a flavonoid based on its adduct ion at *m*/*z* 525 [M−H]^−^ and *m*/*z* 1051 [2M−H]^−^. The molecular formula was confirmed as C_30_H_22_O_9_ using the chemical composition function of the Waters UNIFI program. Referring to previous papers, it was tentatively inferred to be the compound daphnodorin C [17]. Compound 21 was identified as a flavonoid based on the adduct ion at *m*/*z* 525 [M−H]^−^ and *m*/*z* 1051 [2M−H]^−^. The molecular formula was confirmed as C_30_H_22_O_9_ using the chemical composition function of the Waters UNIFI program. Referring to previous papers, it was tentatively inferred to be the compound daphnodorin D2 [18]. The compounds mentioned above are major compounds, and these compounds need to be further confirmed using standard compounds.

### 3.2. DKS Promoted In Vitro Wound Healing in a Dose- and Time-Dependent Manner

Prior to evaluating the migratory effect of DKS, its cytotoxicity was assessed. No significant changes were observed within 24 h; however, cell viability slightly decreased after 48 h (Figure 2A). To examine DKS’s impact on migration, we performed a scratch test. The results indicated that DKS treatment enhanced scratch wound closure (Figure 2B,C). Analysis using ImageJ revealed that 40 µg/mL of DKS significantly enhanced HaCaT cell migration compared to vehicle control after both 24 and 48 h of treatment (Figure 2C). These findings indicate that DKS positively contributes to cell migration. The results from the Transwell migration assay confirmed that DKS promotes HaCaT cell migration at a concentration of 40 µg/mL (Figure 2D). Hence, 40 µg/mL of DKS was selected for subsequent experiments.

### 3.3. DKS-Induced Cell Migration Was Attenuated by MEK1/2 Inhibition

ERK activation is known to play a crucial role in epithelial cell migration, and its inhibition impairs this process [19]. Western blot analysis was performed to determine the phosphorylation level of ERK induced by DKS. HaCaT cells were treated with 40 µg/mL of DKS for 5 and 10 min. Following DKS treatment, the level of phospho-ERK significantly increased within 5 min and was sustained at 10 min (Figure 3A). This ERK phosphorylation was blocked by pretreatment with the selective MEK1/2 inhibitor U0126 (Figure 3B). To examine whether DKS-induced ERK activation contributes to HaCaT cell migration, cells were pretreated with U0126 for 1 h, followed by incubation with DKS for 24 and 48 h. The enhanced cell migration induced by DKS was reversed by U0126 treatment (Figure 3C,D).

### 3.4. DKS-Induced ERK Activation Upregulates MMP9 Expression, and Its Inhibition Reduces Cell Migration

We investigated the expression level of MMP9, as MMPs play a crucial role in keratinocyte migration [20]. DKS treatment significantly upregulated *MMP9* mRNA expression at both 24 and 48 h (Figure 4A). Zymography further confirmed that DKS elevated MMP9 protein levels, which were suppressed by U0126 pretreatment (Figure 4B). To assess the role of MMP9 in DKS-induced migration, a scratch test was performed. The enhanced migration was attenuated by the MMP9 inhibitor (Figure 4C,D). These findings suggest that DKS-mediated cell migration is regulated through the ERK/MMP9 signaling pathway.

### 3.5. DKS Fraction 5 (DKF5) Plays a Major Role in Demonstrating the Migratory Impact of DKS Without Cytotoxicity

Based on previous results showing that DKS promotes cell migration through the ERK/MMP9 pathway, we next aimed to identify which fraction contributes most to this effect. Therefore, six fractions from DKS were evaluated at half the dose of DKS (20 µg/mL) based on the assumption that the fractions may be more potent than the crude extract while minimizing potential cytotoxicity in the context of *MMP9* mRNA upregulation. Among the six groups, DKS fraction 5 (DKF5) exhibited the highest level of *MMP9* mRNA expression (Appendix A). For this reason, subsequent experiments were conducted using DKF5.

To determine the optimal dose of DKF5, we first evaluated *MMP9* mRNA expression across a concentration range from 0 to 20 μg/mL. A statistically significant increase in *MMP9* mRNA levels was observed starting at 1 μg/mL (Appendix A). Subsequently, we assessed the cytotoxicity of DKF5 at varying concentrations (0, 1, 10, and 20 µg/mL). No cytotoxicity was observed at concentrations up to 10 µg/mL, whereas cytotoxic effects were evident at 20 µg/mL (Figure 5A). Subsequently, a scratch test was performed using the same concentration range, which showed a consistent and statistically significant enhancement in cell migration at all tested concentrations compared to the control (0 µg/mL) (Figure 5B,C). Consequently, we selected 1 µg/mL as the optimal concentration for further experiments. This selection was further supported by Transwell migration data, which revealed an increase in cell movement following treatment with 1 µg/mL of DKF5 for 24 h (Figure 5D).

### 3.6. DKF5 Induced ERK Phosphorylation, Leading to Enhanced Cell Migration

We investigated whether DKF5 induces ERK activation, as observed with DKS. Upon exposure of HaCaT cells to 1 µg/mL of DKF5 for 10 min, ERK phosphorylation was significantly increased, and this effect was entirely suppressed in the presence of U0126 (Figure 6A). To determine whether this activation affects keratinocyte migration, a scratch test was performed. The results demonstrated that the DKF5-induced migration was attenuated by U0126 (Figure 6B,C).

### 3.7. DKF5-Induced ERK Activation Regulates MMP9 Expression, with MMP9 Inhibition Partially Reducing Cell Migration

To assess whether DKF5 induces MMP9 expression via ERK activation, HaCaT cells were exposed to 1 µg/mL of DKF5 for 24 h with or without pre-treatment with U0126 for 1 h. The results confirmed that DKF5 up-regulates *MMP9* mRNA expression, an effect that was suppressed by U0126 (Figure 7A). Furthermore, zymography analysis revealed increased levels of MMP9 expression induced by DKF5 treatment, which was attenuated by co-treatment with U0126 (Figure 7B). A scratch test further showed that DKF5-induced migration was partially inhibited by the MMP9 inhibitor, highlighting the involvement of MMP9 in this process (Figure 7C,D).

## 4. Discussion

Various natural products have been explored for their potential to enhance wound healing [21]. For example, curcumin has been widely studied for its wound-healing properties. However, the further application of curcumin is limited due to its rapid metabolism and poor biocompatibility in human physiology [22]. In our study, we identified a crude extract of DKS that promotes keratinocyte migration, an essential factor in wound closure. Our findings suggest that fractionated DKS (DKF5) exhibits significant wound-healing potential at a lower, non-cytotoxic dose. DKF5 is a multi-compound formulation composed of flavonoids, a class of phenolic compounds abundantly found in fruits, vegetables, herbs, and other plants. Flavonoids are among the most prevalent bioactive compounds and have been extensively studied for their therapeutic potential. Notably, natural flavonoids have been reported to exert beneficial effects on wound healing in both in vitro and in vivo models [23,24]. These results mark the therapeutic potential of DKS-derived compounds in promoting migration.

One of the key findings of this study is the identification of a fractionated natural product, DKF5, as a potent wound-healing agent. We initially tested the crude extract of DKS and observed a dose-dependent effect on keratinocyte migration; however, at 40 µg/mL, cytotoxicity was evident after 48 h (Figure 2A). To mitigate this issue, we proceeded to investigate DKS fractions and identified DKF5, which exhibited comparable migration-enhancing properties at a significantly lower dose of 1 µg/mL without cytotoxic effects. These results suggest that DKF5 may serve as a safer and more effective agent to promote epithelial cell migration while minimizing cytotoxicity-related concerns.

Our results reveal that DKS and its fraction DKF5 modulate the ERK-MMP9 pathway, enhancing keratinocyte migration primarily through collective movement. ERK activation has been widely recognized for its role in keratinocyte migration and wound healing [19,25]. In our study, DKF5 treatment significantly increased ERK phosphorylation levels (Figure 6A), which correlates with enhanced keratinocyte motility. Furthermore, we observed an increase in MMP9 expression following DKF5 treatment, similar to its role in mouse wound healing models where MMP9 is upregulated before wound closure [26]. Importantly, MMP9 knock-out models exhibit delayed healing [27]. Thus, the upregulation of MMP9 following DKF5 treatment suggests a potential role in ECM remodeling through ERK-dependent signaling.

While conventional approaches to studying cell migration typically focus on the epithelial-to-mesenchymal transition (EMT), our study highlights the collective migration of HaCaT cells, which are human-derived epithelial keratinocytes. Our findings suggest that DKS-induced keratinocyte migration does not fully align with the conventional EMT in wound healing. Conventional EMT is characterized by a decrease in E-cadherin and an increase in Vimentin, Slug, and N-cadherin, with TGFβ1 levels often being altered in response to extracellular stimuli. While our results initially appear to align with conventional EMT due to the increase in Vimentin and Slug, the extent of Slug upregulation was relatively modest, and, notably, N-cadherin exhibited a significant decrease rather than an increase. Furthermore, despite DKS-induced cellular changes, TGFβ1, which is known to drive EMT, was not upregulated but instead showed a marked reduction, further highlighting how our findings differ from the classical EMT framework (Appendix A, Appendix A). Furthermore, previous reports have shown that in an in vitro wound model, epithelial monolayers predominantly exhibit collective migration [28]. During this process, ERK activation propagates throughout epithelial cell sheets [19]. However, EMT is observed in primary skin cells [29]. Thus, EMT and collective migration are not mutually exclusive but instead represent complementary aspects of the migratory behavior of epithelial cells [30].

Inflammation is the first stage of wound healing, and its timely initiation is critical for accelerating the overall healing process. Ideally, the inflammatory phase should begin within 24 h of post-injury, as a prompt response can facilitate the initiation of subsequent wound healing stages. In this context, our experimental results demonstrate that DKS crude extract significantly increased the mRNA expression of *IL-1β*, *IL-8*, and *TNF-α* at 24 h post-treatment (Appendix A, Appendix A), suggesting its potential role in initiating the early inflammatory response. A similar effect was observed with DKF5, further supporting its pro-inflammatory activity (Appendix A, Appendix A). Notably, the increase in *IL-8* mRNA levels is particularly significant because a previous study has reported that *IL-8* levels were elevated during the accelerated collective migration of HaCaT cells [31], suggesting that its upregulation may contribute to the early stages of wound healing by promoting keratinocyte migration. Furthermore, the fact that U0126 attenuated the DKF5-induced increase in pro-inflammatory cytokines, including *IL-8* mRNA levels, supports the involvement of the ERK pathway in this process (Appendix A, Appendix A). These findings indicate that the modulation of early inflammatory signaling through the ERK activation may contribute to the wound healing process, highlighting the contribution of DKS and DKF5 to early inflammatory signaling. More broadly, they suggest that the early induction of inflammatory cytokines not only accelerates the onset of wound healing but also plays a crucial role in promoting keratinocyte migration, thereby enhancing tissue regeneration.

While our findings highlight the therapeutic potential of DKF5, some limitations remain. A key limitation of our study is its reliance on in vitro models, which may not fully replicate the complexity of in vivo wound healing. Although our findings suggest potential therapeutic applications of DKS for wound healing, further investigations, particularly in animal models and eventually clinical trials, are needed to validate their efficacy and safety. Moreover, conducting additional mechanistic studies could offer a more comprehensive understanding of DKS’s mode of action and its potential as a therapeutic agent for wound healing.

In summary, this study elucidates the intricate mechanisms underlying the wound-healing properties of DKS and its fraction DKF5, highlighting the integral roles of ERK signaling and MMP9 induction in promoting cell migration. These insights not only enhance our understanding of wound healing processes facilitated by DKS and DKF5 but also offer potential for the development of novel therapeutic strategies to promote wound recovery.

## 5. Conclusions

The crude extract of DKS demonstrated notable effects on keratinocyte migration, suggesting its potential as a wound-healing agent. Our findings further highlight the therapeutic potential of DKF5, a fraction derived from DKS, in promoting keratinocyte migration at a non-cytotoxic concentration. These results indicate that DKF5 may serve as a viable candidate for further therapeutic development. However, while our study provides foundational evidence, further investigations are required to elucidate the precise mechanisms underlying its bioactivity and assess its in vivo efficacy. Overall, our research contributes to the growing body of evidence supporting natural product-based interventions for enhanced wound healing, supporting the development of novel plant-based therapeutic strategies.

## Figures and Tables

**Figure 1 cimb-47-00300-f001:**
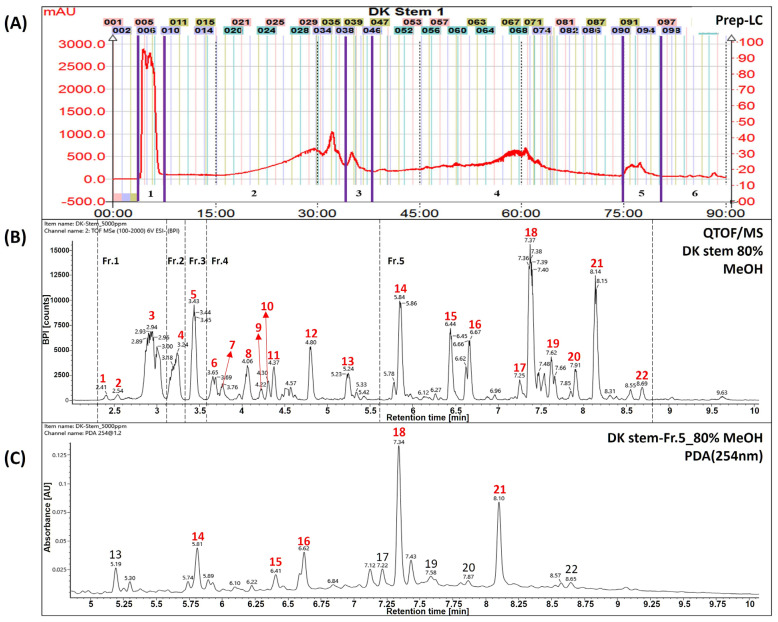
Identification of major compounds in *Daphne kiusiana* stem (DKS) extract and its fraction number 5. (**A**) Separation chromatogram of the stem extract of *Daphne kiusiana* using prep-LC. (**B**) Base peak intensity chromatograms of DKS using UPLC-QToF/MS in negative mode. (**C**) UPLC-PDA chromatograms of fraction No. 5 of the stem extract from *Daphne kiusiana* (DKF5). In (**C**), the red-labeled numbers indicate the five principal components of fraction number 5.

**Figure 2 cimb-47-00300-f002:**
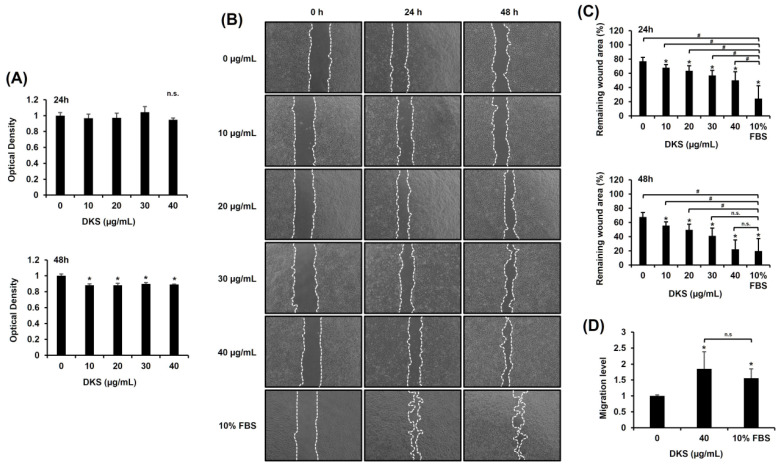
*Daphne kiusiana* stem extracts enhance epithelial cell migration. HaCaT cells were exposed to varying concentrations of *Daphne kiusiana* stem extract (DKS). (**A**) Relative cell viability was assessed using the EZ-Cytox kit, with DMSO-treated control (0 µg/mL) cells serving as the reference for 100% viability. (**B**,**C**) Following 2 h of incubation with mitomycin C (5 µM), monolayers of cells were subjected to scratching and then treated with the specified concentrations of DKS. Images of the scratched wound were taken at 0, 24, and 48 h at 200× magnification, followed by quantification. (**D**) Following a 24-h exposure to DKS, the number of migrated cells was quantified by measuring the number of cells in the lower membrane of the Transwell insert. The presented data represent the mean ± SD (*n* = 3). Statistical significance is denoted by * *p* < 0.05 compared to control (0 µg/mL) and # *p* < 0.05 compared to the 10% FBS-treated group. (**A**,**D**) were analyzed using a paired student *t*-test, while (**C**) underwent one-way ANOVA with Tukey post hoc analysis. (n.s.: not significant).

**Figure 3 cimb-47-00300-f003:**
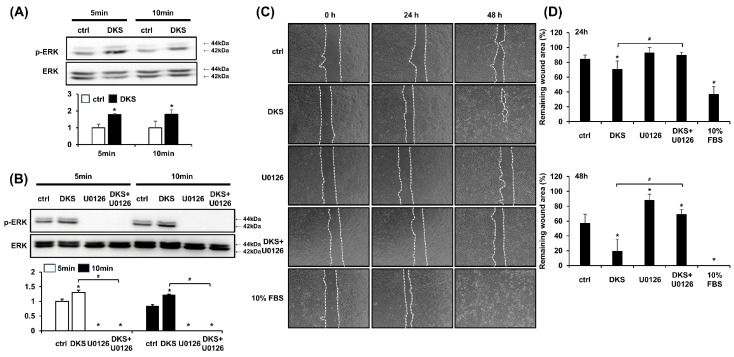
*Daphne kiusiana* stem extract-induced HaCaT cell migration depends on ERK activation. (**A**,**B**) Cells were exposed to 40 µg/mL of *Daphne kiusiana* stem extract (DKS) for the specified periods in the presence or absence of U0126 (10 µM). ERK activation was subsequently analyzed by western blot. (**C**,**D**) Following 2 h of incubation with mitomycin C (5 µM), monolayers of cells were subjected to scratching. Then, the cells were treated with DKS in the presence or absence of U0126. Images of the scratched wound were taken at 0, 24, and 48 h at 200× magnification, followed by quantification. The presented data represent the mean ± SD (*n* = 3). Statistical significance is denoted by * *p* < 0.05 compared to control (ctrl) and # *p* < 0.05 compared to the DKS-treated group. (**A**) was analyzed using a paired student *t*-test, while (**B**,**D**) underwent one-way ANOVA with Tukey post hoc analysis.

**Figure 4 cimb-47-00300-f004:**
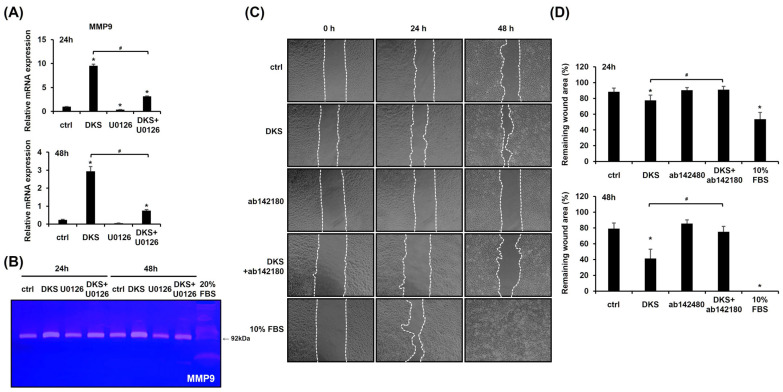
Cell migration induced by *Daphne kiusiana* stem extracts occurs through the ERK/MMP9 pathway. (**A**) HaCaT cells were exposed to 40 µg/mL of *Daphne kiusiana* stem extract (DKS) for the indicated periods in the presence or absence of U0126 (10 µM). *MMP9* gene expression was assessed using qRT-PCR, with *GAPDH* serving as the reference gene. (**B**) Using the same conditions, zymography was performed to confirm the expression level of the MMP9 protein. Here, 20% FBS-treated cells served as a positive control because FBS enhanced HaCaT cell migration. (**C**,**D**) Following 2 h of incubation with mitomycin C (5 µM), monolayers of cells were subjected to scratching. Then, the cells were treated with DKS (40 µg/mL) in the presence or absence of the MMP9 inhibitor ab142180 (10 µM). Images of the scratched wound were taken at 0, 24, and 48 h at 200× magnification, followed by quantification. The presented data represent the mean ± SD (*n* = 3). Statistical significance is denoted by * *p* < 0.05 compared to control (ctrl) and # *p* < 0.05 compared to the DKS-treated group.

**Figure 5 cimb-47-00300-f005:**
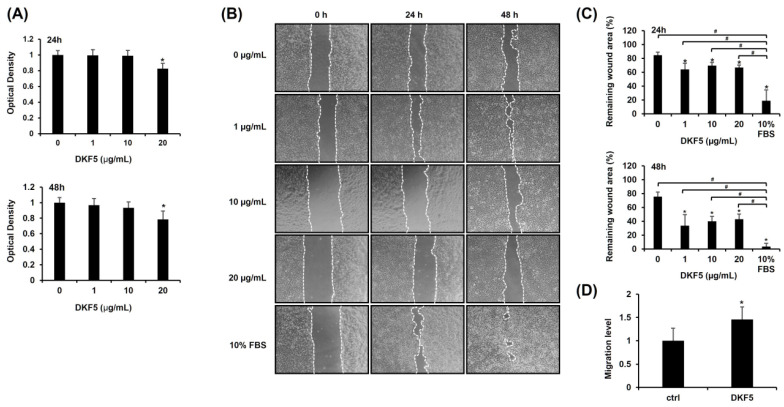
Fraction No. 5 of stem extract from *Daphne kiusiana* (DKF5) enhances HaCaT cells. (**A**) HaCaT cells were exposed to varying concentrations of DKF5 for 24 and 48 h. Relative cell viability was assessed using the EZ-Cytox kit, with DMSO-treated cells (0 µg/mL) serving as the reference for 100% viability. (**B**,**C**) Following 2 h of incubation with mitomycin C (5 µM), monolayers of cells were subjected to scratching and then treated with the specified concentrations of DKF5. Images of the scratched wound were taken at 0, 24, and 48 h at 200× magnification, followed by quantification. (**D**) Following a 24-h exposure to DKF5 (1 µg/mL), the migration capability was quantified by measuring the number of cells in the lower membrane of the Transwell insert. The presented data represent the mean ± SD (*n* = 3). Statistical significance is denoted by * *p* < 0.05 compared to control (0 µg/mL) and # *p* < 0.05 compared to 10% FBS-treated group. (**A**,**D**) were analyzed using a paired student *t*-test, while (**C**) was analyzed by one-way ANOVA with Tucky post hoc analysis.

**Figure 6 cimb-47-00300-f006:**
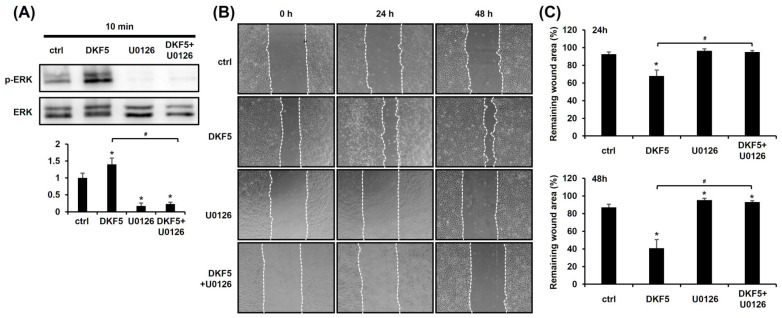
Fraction No. 5 of stem extract from *Daphne kiusiana* (DKF5)-induced cell migration depends on ERK activation. (**A**) HaCaT cells were exposed to DKF5 (1 ug/mL) in the presence or absence of U0126 (10 µM) for 10 min. Then, cell lysates were prepared, followed by western blot analyses. (**B**,**C**) Following 2 h of incubation with mitomycin C (5 µM), monolayers of cells were subjected to scratching. Then, cells were exposed to DKF5 (1 µg/mL) in the presence or absence of U0126 (10 µM). Images of the scratched area were carried out at 0, 24, and 48 h at 200× magnification, followed by quantification. The presented data represent the mean ± SD (*n* = 3). Statistical significance is denoted by * *p* < 0.05 compared to control (ctrl) and # *p* < 0.05 compared to the DKF5-treated group. All data were analyzed using one-way ANOVA with Tukey post hoc analysis.

**Figure 7 cimb-47-00300-f007:**
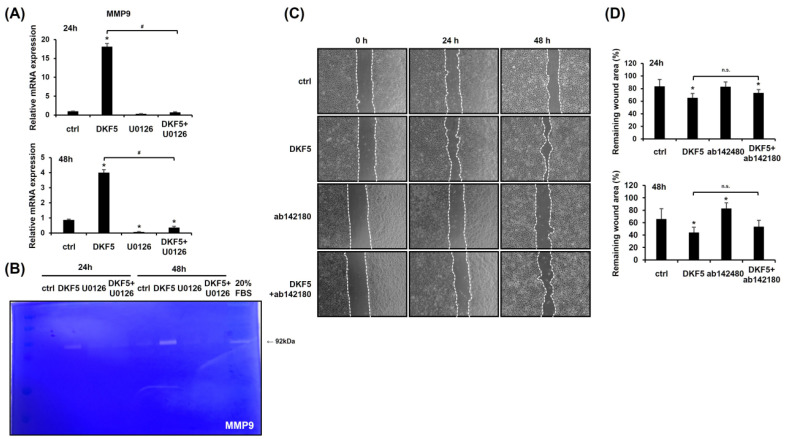
HaCaT cell migration stimulated by fraction No. 5 of stem extract from *Daphne kiusiana* (DKF5) proceeds via the ERK/MMP9 signaling pathway. (**A**) HaCaT cells were exposed to DKF5 (1 µg/mL) in the presence or absence of U0126 (10 µM) for 24 and 48 h, and the expression of the *MMP9* gene was analyzed using qRT-PCR, with *GAPDH* serving as the reference gene. (**B**) Employing the same conditions, zymography was conducted to validate the expression level of the MMP9 protein. (**C**,**D**) Following 2 h of incubation with mitomycin C (5 µM), monolayers of cells were subjected to scratching. Then, the cells were treated with DKF5 (1 µg/mL) in the presence or absence of the MMP9 inhibitor ab142180 (10 µM). Images of the scratched wound were taken at 0, 24, and 48 h at 200× magnification, followed by quantification. The presented data represent the mean ± SD (*n* = 3). Statistical significance is denoted by * *p* < 0.05 compared to control (ctrl) and # *p* < 0.05 compared to the DKF5-treated group. All data were analyzed using one-way ANOVA with Tukey post hoc analysis. (n.s.: not significant).

**Table 1 cimb-47-00300-t001:** The *in-house* library of secondary metabolites (**1**–**22**) obtained using UPLC-QToF/MS for *Daphne kiusiana* stem.

No.	RT (min)	Tentative Compound	Formula	Detected *m*/*z*	Calculated *m*/*z*	MS Error	MS Fragment	Adduct	Ref.
**1**	2.40	Daphnin isomer	C_15_H_16_O_9_	339.0711	339.0721	0.83	177	-H	[15]
**2**	2.54	5-Hydroxy-7-methoxy-8-*O*-*β*-D-glucosylcoumarin	C_16_H_18_O_10_	369.0831	369.0827	−0.59	177	-H	[15]
**3**	2.94	Daphnetin 8-*O*-*β*-D-glucopyranoside	C_15_H_16_O_9_	339.0711	339.0721	0.15	177	-H	[15]
**4**	3.24	Unknown	C_21_H_26_O_13_	485.1292	485.1300	−1.09	177	-H	-
**5**	3.43	Daphnin	C_15_H_16_O_9_	339.0711	339.0721	−0.22	177	-H	[15]
**6**	3.65	5-*O*-*β*-D-Glucosyl-7-methoxy-8-hydroxycoumarin	C_16_H_18_O_10_	369.0831	369.0827	−0.27	192	-H	[15]
**7**	3.76	7,8-Dihydroxycoumarin (=daphnetin)	C_9_H_6_O_4_	177.0201	177.0193	2.29	-	-H	[15]
**8**	4.06	Unknown	C_33_H_44_O_18_	727.2454	727.2454	−1.91	-	-H	-
**9**	4.22	Unknown	C_16_H_20_O_10_	371.0992	371.0983	0.26	177	-H	-
**10**	4.30	Unknown	C_28_H_38_O_15_	613.2125	613.2137	−1.09	579, 471, 287, 177	-H	-
**11**	4.37	Unknown	C_35_H_48_O_20_	787.2633	787.2666	−1.91	579, 417, 181	-H	-
**12**	4.80	Unknown	C_26_H_34_O_11_	521.2013	521.2028	−0.56	329, 285	-H	-
**13**	5.24	Unknown	C_26_H_32_O_11_	519.1873	519.1871	−0.58	357, 177	-H	-
**14 ***	5.84	Daphnodorin I	C_30_H_22_O_10_	541.1131	541.1140	−1.71	371	-H	[14]
**15 ***	6.44	Nortrachelogenin	C_20_H_22_O_7_	373.1281	373.1292	−0.44	-	-H	[15]
**16 ***	6.67	Daphnodorin B	C_30_H_22_O_10_	541.1131	541.1140	−1.71	351, 336, 151	-H	[16]
**17**	7.25	Daphnodorin A	C_30_H_22_O_9_	525.1189	525.1191	−1.32	431, 253	-H	[17]
**18 ***	7.37	Daphnodorin C	C_30_H_22_O_9_	525.1189	525.1191	−1.60	295	-H	[17]
**19**	7.62	Daphnodorin F	C_30_H_22_O_9_	541.1131	541.1140	−1.34	523, 281, 151	-H	[18]
**20**	7.91	Daphnodorin E	C_30_H_22_O_10_	541.1131	541.1140	−0.92	523, 389, 151	-H	[18]
**21 ***	8.14	Daphnodorin D2	C_30_H_22_O_9_	525.1189	525.1140	−1.39	295	-H	[18]
**22**	8.69	Daphneolone	C_17_H_18_O_3_	269.1190	269.1183	−0.14	-	-H	[17]

The asterisk mark (*) indicates the major compounds of DKS No. 5 in the UV 254 nm chromatogram.

## Data Availability

All data produced or analyzed in this study are included in the published article. Any additional data or relevant information can be provided by the corresponding author upon request.

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
