# Peer review of "Daphne kiusiana Crude Extract and Its Fraction Enhance Keratinocyte Migration via the ERK/MMP9 Pathway"

_cimb, 2025, doi:10.3390/cimb47050300_

Round 1

Reviewer 1 Report

Comments and Suggestions for Authors

The manuscript by Jinu Lee et al. desired Daphne kiusiana crude extract and its fraction enhanced keratinocyte migration via ERK/MMP9 pathway. The results indicated that Daphne Kiusiana significantly enhanced keratinocyte migration without inducing notable cytotoxicity at tested concentrations, and this effect was mediated through the modulation of matrix metalloproteinase-9 (MMP-9) via extracellular signal-regulated kinase (ERK) signaling. The research suggests Daphne Kiusiana may serve as a potential therapeutic agent for enhancing keratinocyte migration in wound healing. However, there are many issues need to be discussed. The specific problems are as follows:

  1. How to separate the component DKF5?
  1. What is the basis for selecting drug dosage? DKS (40 µg/m) or DKF5 (1 µg/ml)
  1. Have the main compounds not been identified in other components except for DKS5? Why only choose DKS5 for in vitro experiments instead of separating each component and conducting in vitro experiments together?
  1. The data in Figures is completely inconsistent with the description. The article may be missing some charts.
  1. Figure 3A does not indicate which substance's mRNA it is. In Figure 3B, where does 20% FBS come from? And the MMP9 of the 20% FBS group has not yet been released. So this data is problematic.

Author Response

We are pleased to resubmit our revised manuscript entitled "Daphne kiusiana crude extract and its fraction enhanced keratinocyte migration via ERK/MMP9 pathway" for further consideration in Current Issues in Molecular Biology.

The insightful comments and constructive feedback provided by the reviewers and the editorial team are sincerely appreciated. In response, we have carefully addressed all the points raised during the review process. The manuscript has been thoroughly revised to improve clarity, logical flow, and scientific rigor. Furthermore, we have carefully refined the language throughout the text to improve its linguistic accuracy and academic tone.

We would also like to extend our apologies for any inconvenience caused by the original organization of the manuscript. In this revision, we have made substantial efforts to improve the structure and presentation of the content to ensure a clearer and more coherent narrative.

With these revisions, we believe that the revised version of our manuscript addresses all concerns and significantly strengthens the overall quality of the submission. We are grateful for the opportunity to revise our work and hope it now meets the standards required for publication in Current Issues in Molecular Biology.

Please see the attachment for the reponse.

Thank you for your continued consideration. Please feel free to contact us if you require any additional information.

Sincerely,

Jeong-Hwa Baek, D.D.S., Ph.D.

Department of Molecular Genetics,

School of Dentistry and Dental Research Institute,

Seoul National University,

Seoul 08826,

Republic of Korea

Reviewer 2 Report

Comments and Suggestions for Authors

The topic of this manuscript is interesting, and there is potential for acceptance in this journal. However, the manuscript is currently not well-organized, and I believe it should be revised and resubmitted after improving the overall formatting and structure. A revised version addressing the following points would be eligible for further review.

The figure numbers cited in the main text do not match the actual figures, making it impossible to verify the results. This is the main reason for the rejection.

Below are some additional minor comments:

  • Line 109: Is “TB Green Premix EC Tag” a typo for “Taq”?

  • Line 113: Please clarify that the HaCaT cells are of human origin.

  • Line 118: “DKS (40 µg/m)” — Should this be “µg/mL”?

  • Line 119: Please provide the manufacturers of U0126 and ab142180. Note that “ab142180” is a catalog number; please also provide the full name of the reagent. The authors should describe the function of the reagent, such as stating that it is an “xxx inhibitor”.

Author Response

(The authors gave the same response as above.)

Round 2

Reviewer 1 Report

Comments and Suggestions for Authors

The author has basically answered my questions.

Reviewer 2 Report

Comments and Suggestions for Authors

The authors modified the manuscript appropriately.

So, this paper is acceptable in CIMB.